# Modeling the Physiological Parameters of Brewer's Yeast during Storage with Natural Zeolite-Containing Tuffs Using Artificial Neural Networks

Anton V. Shafrai [1,*], Larisa V. Permyakova [2], Dmitriy M. Borodulin [1] and Irina Y. Sergeeva [2]

1 Institute of Engineering Technologies, Kemerovo State University, 6, Krasnaya St., 650000 Kemerovo, Russia
2 Technological Institute of Food Industry, Kemerovo State University, 6, Krasnaya St., 650000 Kemerovo, Russia
* Correspondence: shafraia@mail.ru; Tel.: +7-903-984-3846

**Abstract:** Various methods are used to prevent the deterioration of the biotechnological properties of brewer's yeast during storage. This paper studied the use of artificial neural networks for the mathematical modeling of correcting the biosynthetic activity of brewer's seed yeast of the C34 race during storage with natural minerals. The input parameters for the artificial neural networks were the suspending medium (water, beer wort, or young beer); the type of the zeolite-containing tuff from Siberian deposits; the tuff content (0.5–4% of the total volume of the suspension); and the duration of storage (3 days). The output parameters were the number of yeast cells with glycogen, budding cells, and dead cells. In the yeast stored with tuffs, the number of budding cells increased by 1.2–2.5 times, and the number of cells with glycogen increased by 9–190% compared to the control sample (without tuff). The presence of kholinskiy zeolite and shivyrtuin tuffs resulted in a significant effect. The artificial neural networks were required for solving the regression tasks and predicting the output parameters based on the input parameters. Four networks were created: ANN1 (mean relative error = 4.869%) modeled the values of all the output parameters; ANN2 (MRE = 1.8381%) modeled the number of cells with glycogen; ANN3 (MRE = 6.2905%) modeled the number of budding cells; and ANN4 (MRE = 4.2191%) modeled the number of dead cells. The optimal parameters for yeast storage were then determined. As a result, the possibility of using ANNs for mathematical modeling of undesired deviations in the physiological parameters of brewer's seed yeast during storage with natural minerals was proven.

**Keywords:** brewer's yeast; natural zeolite-containing tuffs; storage; physiological parameters; mathematical modeling; artificial neural networks

## 1. Introduction

The yeast culture used in beer production has a significant impact on the course of wort fermentation, and the analytical and organoleptic parameters of the finished drink. The availability of proper quality raw materials does not guarantee obtaining high-quality beer if a yeast of poor viability and activity is used. One of the production stages where deviation from optimal parameters may occur is that of brewer's seed yeast storage before introduction into the fermentation medium. Deviations from the optimal storage duration parameters, optimal temperature, and composition of the incubation medium pose a serious risk to the viability of the yeast population [1–4].

Among the methods and means for eliminating or moderating the negative effect of various factors on the physiological and biochemical characteristics of the microbial mass, the use of preparations and supplements to correct the mineral, nitrogenous, and vitamin composition of the incubation medium can be distinguished [5].

In the practical activity of breweries, the lack of certain mineral components in wort is often compensated for by adding yeast nutrients [5,6]. In most nutrients, mineral

components are present in the inorganic form, which can negatively affect the hygienic safety of the product.

In a number of works, the possibility of modifying the composition of the culture medium by introducing mineral components from natural sources has been investigated. Researchers have proposed using blown sand [7], geothermal water [8], organometallic compounds of magnesium, manganese, iron, zinc, copper, selenium, and organic form of silicon [9,10].

Natural zeolite-containing tuffs (ZT) are widely known, mainly as ion-exchange and sorption materials [11–13]. Furthermore, they have a certain biological activity. In the Russian Federation, the deposits of natural ZT that are promising from the point of view of industrial development are concentrated in Siberia: in the Chita and Kemerovo regions (the Shivyrtuy and the Pegas River deposits respectively), the Republic of Buryatia (the Kholinsk deposit), and the Republic of Sakha (the Khonguruu deposit). Clinoptilolite and heulandite are most common in the tuffs of these deposits, while the presence of perlite and montmorillonite is less common [14,15].

Clinoptilolite with a "loose" crystalline structure is characterized by high biological activity. Montmorillonite and easily soluble compounds with a "mobile" layered structure also have high biological activity. In most cases, these minerals are hydrated; they have mobile exchange ions that easily pass into solution and take part in the processes of the vital activity of a living organism.

Our study focused on researching the possibility of minimizing undesirable deviations in the physiological and biochemical properties of brewer's yeast culture at the storage stage before introduction into the fermented wort by adding natural zeolite-containing tuffs from various Siberian deposits to the incubation medium with further modeling of this process.

Modern methods of mathematical modeling are very diverse and differ in the complexity and accuracy of the resulting models. Artificial neural networks (ANNs) have been used for modeling various processes for a long time, however, they have rarely been used in food science technologies. The number of such studies has been growing every year recently, and ANNs are used to solve regression and classification problems [16–31].

This work aimed to apply ANNs for the mathematical modeling of correcting the biotechnological properties of brewer's yeast at the seed storage stage using natural minerals.

The objectives of the work were:

- To design the ANNs to predict the output values of the process;
- To determine the accuracy of the ANNs;
- Process optimization.

## 2. Material and Methods

### 2.1. Experiment

The research team studied the bottom fermentation yeast *Saccharomyces cerevisiae* C34 of the 4th generation, fermented wort, and young beer obtained directly from a brewing plant (Kemerovo, Russia). Pale malt produced by "Grainrus-Kursk Malt, LLC" (Kursk, Russia) was used to prepare the wort, its properties were: moisture—4.6%, extractivity—81% to the dry matter, mashing method—single mash. The wort was hopped with *Perle* hops (Germany) characterized by: $\alpha$-acid content—6.2%, humidity—7.5%. The properties of the obtained wort met the following brewing standards [2,4]: the dry content fraction—11%, the fermentable sugar content—8.76 g/100 cm$^3$, amine nitrogen concentration—25.60 mg/100 cm$^3$, pH—5.4.

Tuffs from the following deposits were used as biostimulators: kholinskiy zeolite (the Kholinsk deposit, Republic of Buryatia, Russia), shivyrtuin (the Shivyrtuy deposit, Chita region, Russia), and pegasin (the Pegas River deposit, Kemerovo region, Russia). The preparation of zeolites consisted of dust removal by washing with water, subsequent drying at a temperature of 120 °C, and grinding into particles of 50–140 microns in size.

To study the changes in the physiological state of yeast culture at the storage stage in the presence of minerals, the experiment was set as follows. Seed culture yeast was mixed

with a medium (water, 11% hopped beer wort, or young beer with a dry matter content of 2.9% at a 1:1 ratio. ZT was introduced into the storage medium (0.5–4.0% of the suspension volume). The yeast mass was incubated for three days at a temperature of 2–4 °C. The choice of storage parameters (suspension medium, duration, temperature) was determined by the conditions accepted in beer production [1,2,4]. A yeast culture in a ZT-free medium served as a control sample.

### 2.2. Parameter Measurement Procedure

The physiological parameters (the total number of cells; the number of cell with glycogen; the number of budding cells; the number of dead cells) were measured in the original yeast culture and during the storage process using the methods that are common for beer production.

The general yeast concentration was determined by counting with direct microscopy ($600\times$ magnification) in Gorjaev's chamber with Levenhuk 850B binocular microscope (Ningbo Shengheng Optics & Electronics Co., Ltd., Gao Qiao, China).

A budding cell is a cell that does not exceed 1/2 of the parent cell in size. A cell with a bud that is smaller than 1/3 of the parent cell is counted as one cell.

Intravital staining solutions were used to count the number of dead cells and cells with glycogen.

To determine the number of dead cells, a drop in the analyzed yeast culture was mixed with a drop of methylene blue (pH 4.6) and applied to a glass slide. After 2 min, the number of colored cells was counted. The membrane of the dead cells was better penetrated by the stain.

To assess the number of cells with glycogen, a drop of yeast suspension was mixed with 1–2 drops of Lugol's solution and analyzed by microscopy after 2–3 min. The glycogen contained in cells was colored red-brown, the cytoplasm was colored light yellow; the overgrown and weak cells contained no glycogen.

In all of the experiments, the cells were counted in 10 fields, and their number exceeded 600. All of the cells within the grid cells and those crossing the top and right sides of the grid were counted. Before microscopy, the yeast suspension culture was mixed with distilled water for the number of cells in one grid not to exceed 30. The number of dead yeast cells, budding cells, and cells with glycogen was expressed as a percentage of the total number of cells.

The results of the experiment were obtained after 3–4 repetitions. They were expressed in averages with standard deviation (SD). The differences were considered significant if $p \leq 0.05$.

### 2.3. Experimental Data Preparation

Based on the experimental data, a dataset of 204 marked-up records was prepared. The input variables were suspension medium (M), zeolite-containing tuff (T), dose of zeolite-containing tuff (D), and storage duration (SD). The output data were the number of yeast cells with glycogen (CWG), the number of budding yeast cells (BC), and the number of dead yeast cells (DC).

M is a categorical variable that took the values "water", "beer wort", and "young beer". Only quantitative variables were used for the study, so M was replaced with M1, M2, and M3 variables, which took the values 0 or 1 depending on the category. For the "water" value: M1 = 1, M2 = 0, M3 = 0; for the "beer wort" value: M1 = 0, M2 = 1, M3 = 0; for the "young beer" value: M1 = 0, M2 = 0, M3 = 1.

T is a categorical variable that took the values "kholinskiy zeolite", "shivyrtuin", and "pegasin". T was replaced with the T1, T2, and T3 variables, which took the values of 0 or 1 depending on the category. For the "kholinskiy zeolite" value: T1 = 1, T2 = 0, T3 = 0; for the "shivyrtuin" value: T1 = 0, T2 = 1, T3 = 0; for the "pegasin" value: T1 = 0, T2 = 0, T3 = 1.

D took values from 0 to 4% with a step size of 1. SD took values from 0 to 3 days with a step size of 1. The output variables CWG, BC, and DC took values from 0 to 70 (percentage of total).

### 2.4. Research Tools

The ANNs were written in Python in the Google Collaboratory development environment (https://colab.research.google.com accessed on 1 November 2021). The freely distributed PyTorch library was used in this work (https://pytorch.org accessed on 1 November 2021).

### 3. Results

### 3.1. Experimental Results

Suspending the seed yeast in various media with the addition of ZT revealed a positive trend in changing the physiological parameters of the culture during its storage (Figures 1–3). The effectiveness of the mineral from a particular deposit is determined by the dose of ZT added, the composition of the suspension medium, and the duration of contact between the yeast and the storage medium (storage duration).

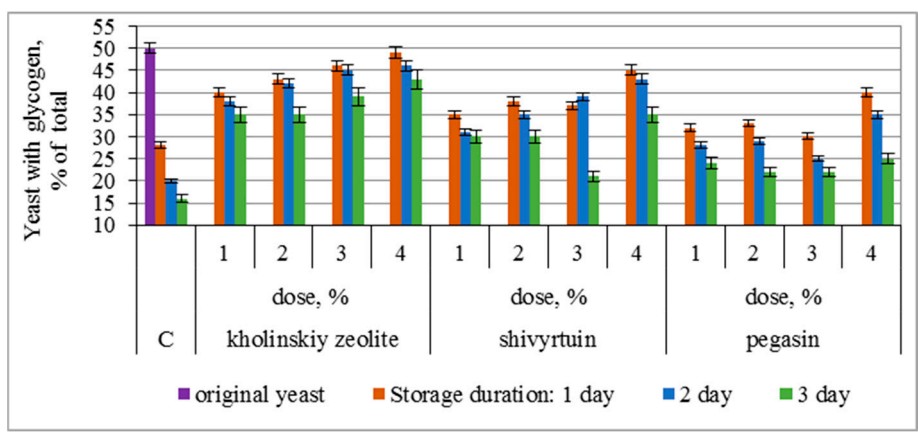

(**a**)

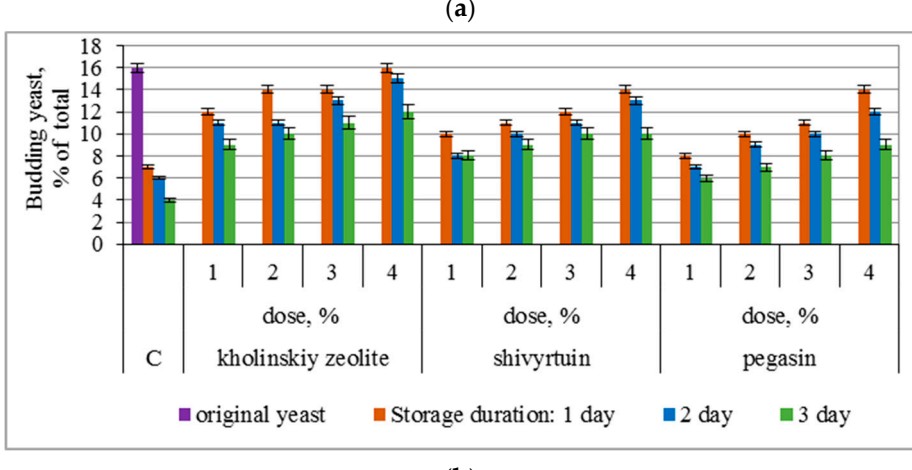

(**b**)

**Figure 1.** *Cont.*

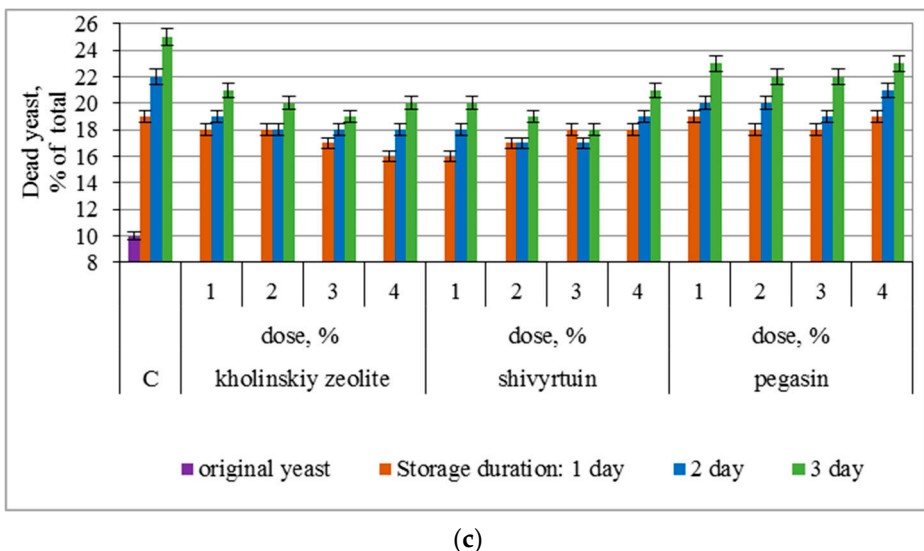

(**c**)

**Figure 1.** Changes in the physiological parameters of yeast culture stored under a layer of water with natural ZT (C—control sample, 1–4% values—dose of zeolite-containing tuffs): (**a**) the number of yeast cells with glycogen; (**b**) the number of budding yeast cells; (**c**) the number of dead yeast cells; ($p \leq 0.05$).

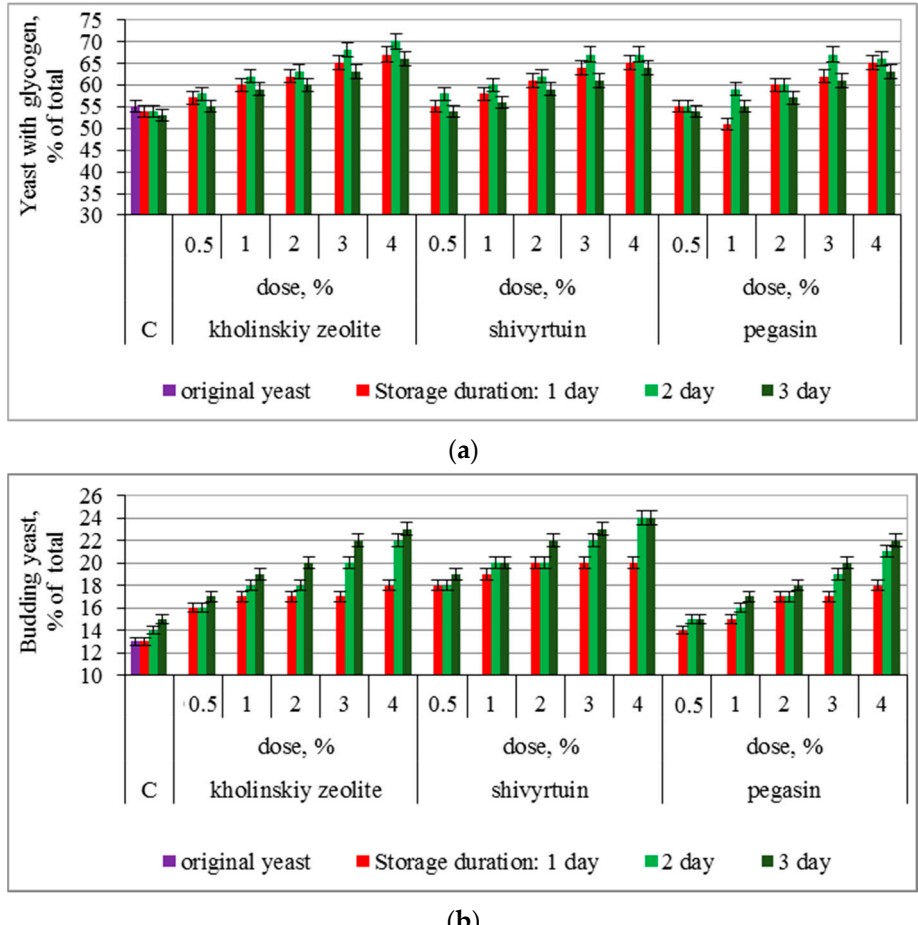

(**a**)

(**b**)

**Figure 2.** *Cont*.

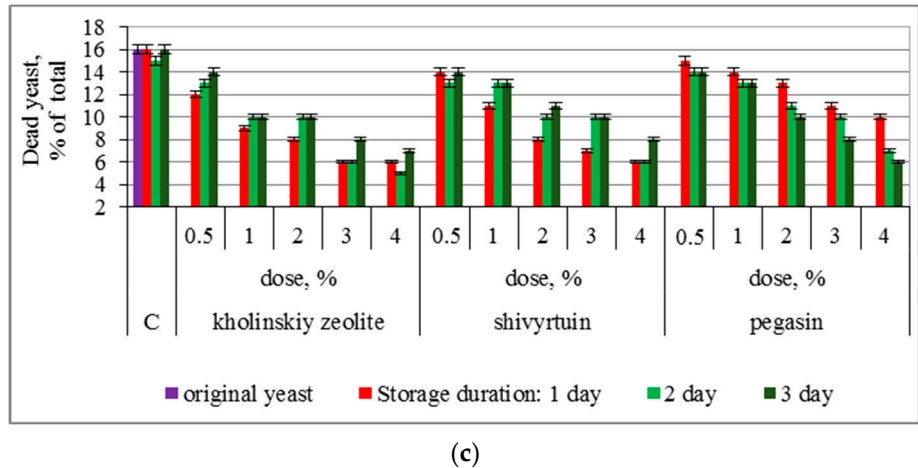

(**c**)

**Figure 2.** Changes in the physiological parameters of yeast culture stored in beer wort with natural ZT (C—control sample, 0.5–4% values—dose of zeolite-containing tuffs): (**a**) the number of yeast cells with glycogen; (**b**) the number of budding yeast cells; (**c**) the number of dead yeast cells; ($p \leq 0.05$).

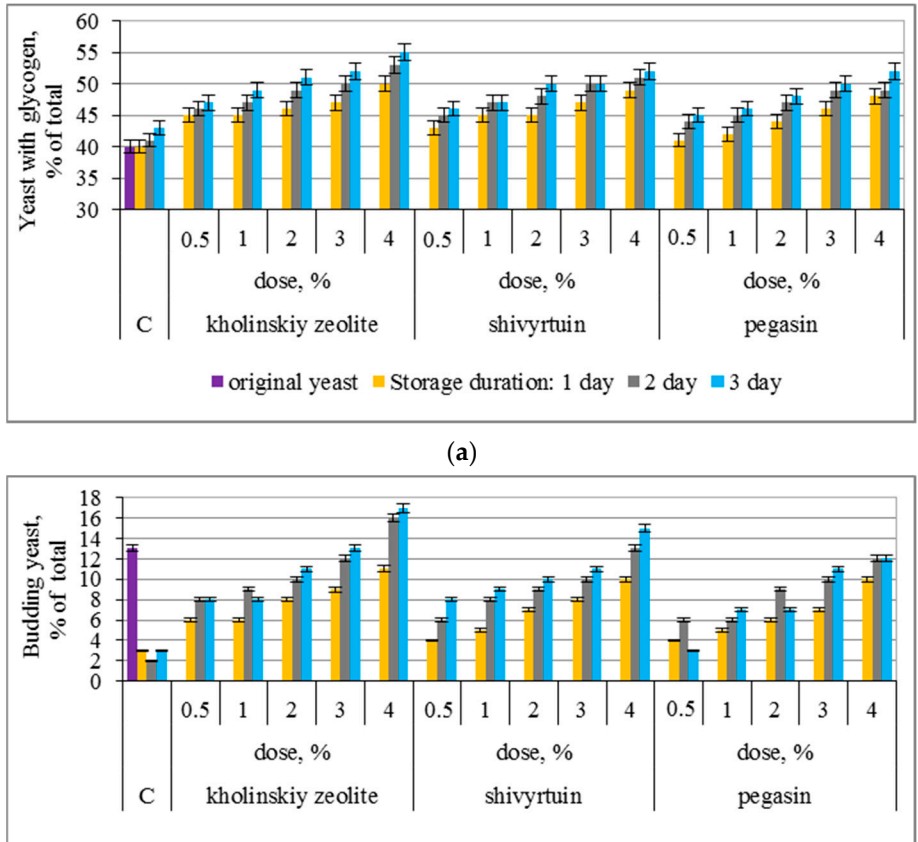

(**a**)

(**b**)

**Figure 3.** *Cont.*

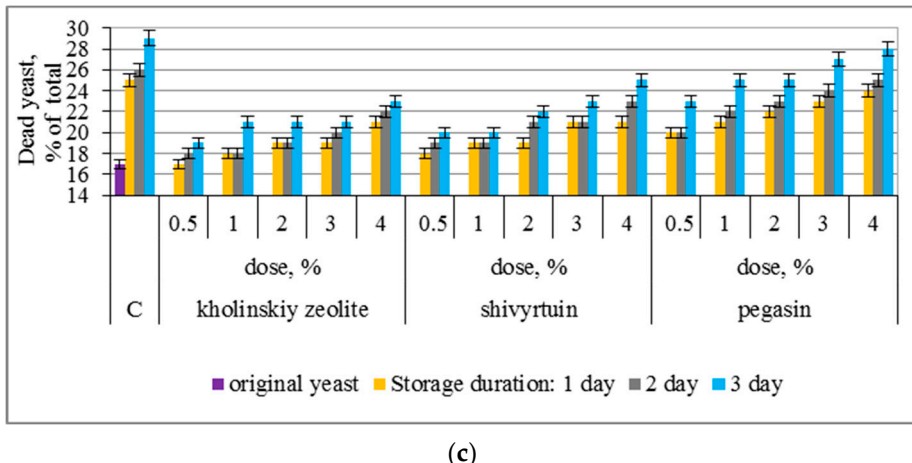

(**c**)

**Figure 3.** Changes in the physiological parameters of yeast culture stored in young beer with natural ZT (C—control sample, 0.5–4% values—dose of zeolite-containing tuffs): (**a**) the number of yeast cells with glycogen; (**b**) the number of budding yeast cells; (**c**) the number of dead yeast cells; ($p \leq 0.05$).

The biosynthetic properties of yeast that has been under a layer of water for a long time are known to deteriorate [2–4,32]. Both in the control sample of yeast stored without introducing ZT into the incubation medium (water) and in the experimental samples, similar negative changes in the studied parameters were observed (Figure 1). However, the presence of minerals in the medium (in the range of 2–4% of the volume) led to an increase in the number of cells with glycogen by 1.9 times on average by the end of the storage process, and a 2.5-time increase in the amount of budding cells in comparison to the control sample. In the experimental samples, the concentration of dead cells was 18% lower than in the control sample.

The effect of tuffs on yeast increases in the case of culture storage in beer wort (Figure 2). After three days of storage, the presence of mineral additives leads to a 9–32% increase in the number of cells with glycogen compared to the initial value, while the number of budding yeast cells increases by 30–85%, and the number of dead cells decreases by 1.2–1.6 times. Over the same time period, the number of cells with glycogen in the control sample (without ZT) increased only by 4%, and the number of budding cells grew by only 15%.

Compared to the experimental series with beer wort, the physiological parameters of the microbial culture stored under a layer of young beer in the presence of ZT demonstrated a similar trend, but at a lower manifestation rate (Figure 3).

In the experimental samples, the number of cells with glycogen and that of the budding cells was 1.2–1.4 times higher than the initial value (on average). The control sample was characterized by an 8% increase in the number of cells with glycogen and a sharp 77% decrease in the number of budding cells compared to the initial value.

Seed yeast storage in young beer with ZT led to an increase in the number of dead cells compared to the initial value, but to a lesser extent than in the control sample without ZT (13% to 64% vs. 71%, respectively).

Regardless of the incubation medium, the presence of kholinskiy zeolite or shivyrtuin provides a more significant increase in the number of budding cells and cells with glycogen compared to the samples with pegasin, with a simultaneous significant reduction in the population of dead cells. The greater the dose of the introduced mineral in all of the media, the more strongly the change in the analyzed parameters of the yeast culture manifests itself.

*3.2. Designing the ANN*

The dataset consisted of eight input and three output parameters. The research hypothesis consisted of designing four ANNs, one of which would predict all three output

parameters, and the other three were to predict one of the three output parameters, respectively. All of the suggested designs were fully-connected ANNs. In the identification process, the following parameters were determined for each of the ANNs:

- The number of hidden layers;
- The number of neurons in the hidden layers;
- The activation function;
- The loss function;
- The step;
- The optimizer;
- Regularization;
- The size and number of batches;
- The number of epochs.

Out of the 204 dataset records, 153 were put in the training set (75%), and 51 were put in the test set (25%).

ANN 1, designed to predict all three output parameters, had three output neurons, each of which showed the modeled CWG, BC, and DC values, respectively. The input values were normalized using the minimum and the maximum values by the formula:

$$x_i' = \frac{x_i - x_{min}}{x_{max} - x_{min}}, \tag{1}$$

where $x'_i$ is the normalized value; $x_i$ is the initial value; $x_{min}$ is the minimum value; and $x_{max}$ is the maximum value.

Thus, all the input values lay within the range from 0 to 1. This normalization showed better results compared to the mean normalization and standard deviation because most input values equaled only 0 or 1 (M1, M2, M3, T1, T2, T3), and the normalization made all the input data homogenous. The output values were not normalized as their normalization showed no positive effect.

ANN 1 comprised two hidden layers with 30 neurons in each; the step was 0.001; the optimizer was *Adam*; the loss function was *MSELoss*; regularization was L2 = 0.00001; the size of batches was 51; the number of batches was 3; the number of epochs was 5269.

The activation function was *GELU* with the formula:

$$GELU(x) = x * \Phi(x), \tag{2}$$

where $x$ is the input value, and $\Phi(x)$ is the cumulative distribution function for Gaussian distribution.

*MSELoss* (mean squared error) was selected as the loss function. It has the formula:

$$MSELoss(y, \hat{y}) = \frac{\sum_{i=1}^{N}(\hat{y_i} - y_i)^2}{N}, \tag{3}$$

where $y$ is the vector of output values from the dataset; $\hat{y}$ is the vector of model output values; and $N$ is the number of data.

The *Adam* (adaptive moment estimation) algorithm was used for optimization. It has the following formula for adjusting weights:

$$W_{t+1} = W_t - \alpha \frac{EMA_{\beta_1}(\nabla f)^t}{\sqrt{EMA_{\beta_2}(\nabla f^2)^t + \varepsilon}}, \tag{4}$$

where $W_{t+1}$ is the new network parameters; $W_t$ is the current network parameters; $\alpha$ is the learning rate; $EMA_{\beta}(\nabla f)^t$ is the exponential moving average of the gradient; $\beta_1$ and $\beta_2$ are the parameters of the exponential moving average; and $\varepsilon$ is the smoothing parameter excluding division by 0.

The learning rate equaled 0.001, and Tikhonov regularization (L2 or weight decay) was used to avoid over-training (its value was 0.00001). ANN 1 was trained in three batches, with 51 records in each. The best results were obtained after 5269 training epochs.

ANN 2 modeled the values for CWG. The input values were normalized in the same way as for ANN 1 using Equation (1); the output values were not normalized.

The architecture of ANN 2 repeats that of ANN 1 with a number of exceptions. The number of hidden layers in ANN2 doubled to make 4, and the number of epochs increased by almost two times to reach 9506.

ANN 3 modeled the values for BC. The normalization of the input and output data was the same as in ANN 1 and ANN 2. ANN 3 comprised four hidden layers with 30 neurons in each; the step was 0.001; the optimizer was Adam; the loss function was MSELoss; regularization was L2 = 0.00001; the size of batches was 51; the number of batches was 3; and the number of epochs was 10,897.

The major feature differing ANN 3 from ANN 2 is the activation function: *Sigmoid*. It is the classical function for the artificial neural network theory and has the formula:

$$Sigmoid(x) = \sigma(x) = \frac{1}{1 + e^{-x}}. \tag{5}$$

ANN 4 modeled the values for DC. For this network, neither the input nor the output values were normalized. ANN 4 comprised six hidden layers with 10 neurons in each; the step was 0.001; the optimizer was Adam; the loss function was MSELoss; regularization was L2 = 0.00001; the size of batches was 51; the number of batches was 3; the number of epochs was 8401; and the activation function was Sigmoid.

## 4. Discussion

### 4.1. Defining the ANN Accuracy

The accuracy of the ANN is an indicator of its quality. There are many accuracy criteria, and the choice of a specific one depends on the task. In the tasks of modeling technological processes, the accuracy of the model is traditionally measured by the difference between the predicted values and the experimental ones, expressed as a percentage. Therefore, in this study, the ANN accuracy was defined using the mean relative error (*MRE*):

$$MRE = \frac{\sum_{i=1}^{N} \frac{|y_i - \hat{y}_i|}{y_i}}{N} * 100, \tag{6}$$

where *MRE* is the mean relative error (%); $y_i$ is the experimental value; $\hat{y}_i$ is the model value; and $N$ is the sample size.

The MRE was calculated for all networks: for ANN 1 it was 4.869% (Figure 4a), for ANN 2 it was 1.8381% (Figure 4b), for ANN 3 it was 6.2905% (Figure 4c), and for ANN 4 it was 4.2191% (Figure 4d). When modeling technological processes, the permissible error is 10%, so the accuracy of the designed ANNs can be considered satisfactory and the models can be used to predict the output variables.

ANN 1 models the values of the three output variables at the same time, while the other networks model one variable each. If we compare their accuracy, it becomes clear that the accuracy of ANN 1 will almost exactly correspond to the arithmetic mean of the accuracies of ANN 2, ANN 3, and ANN 4. Therefore, it is better to use different networks to model the output variables. The values of CWG and DC should be modeled using ANN 2 and ANN 4 because they show greater accuracy for these variables. It is better to model the BC variable with ANN 1 and use only one of its outputs, because this network shows greater accuracy than ANN 3, which was created specifically for BC prediction. A comparison of the experimental and predicted values for each network is shown in Figure 5.

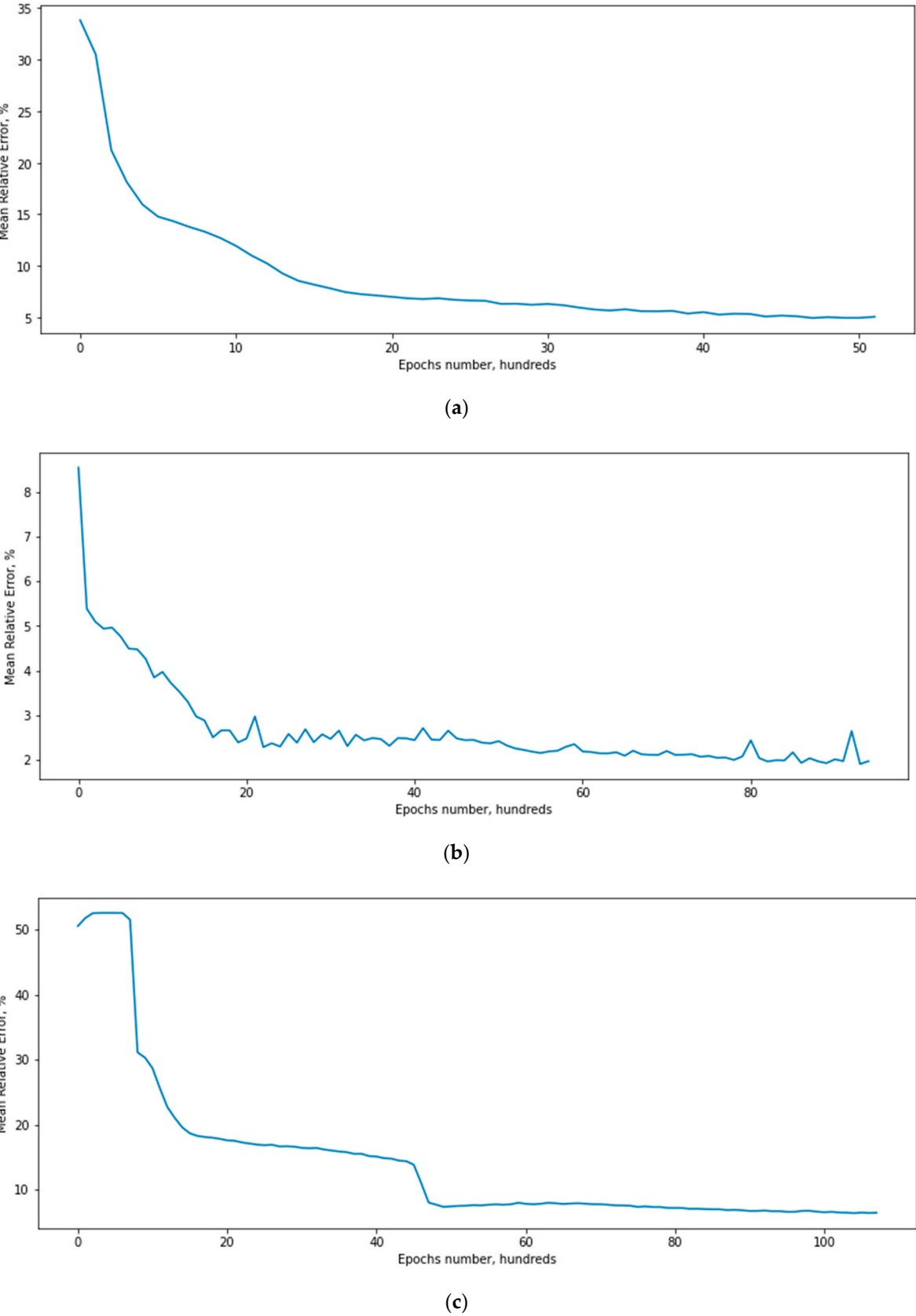

(**a**)

(**b**)

(**c**)

**Figure 4.** *Cont.*

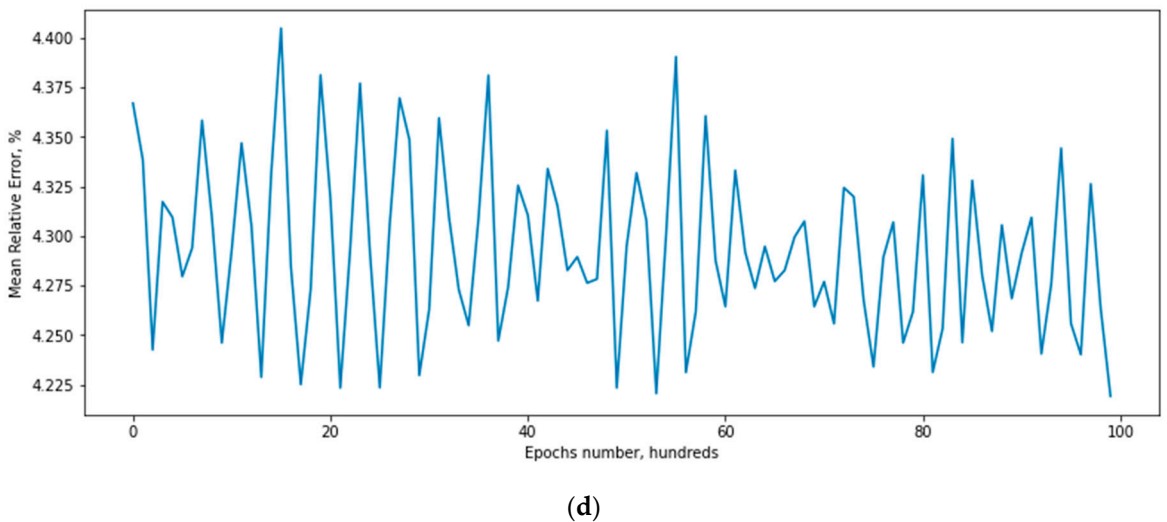

(**d**)

**Figure 4.** (**a**) MRE for ANN 1; (**b**) MRE for ANN 2; (**c**) MRE for ANN 3; (**d**) MRE for ANN 4.

(**a**)

(**b**)

**Figure 5.** *Cont.*

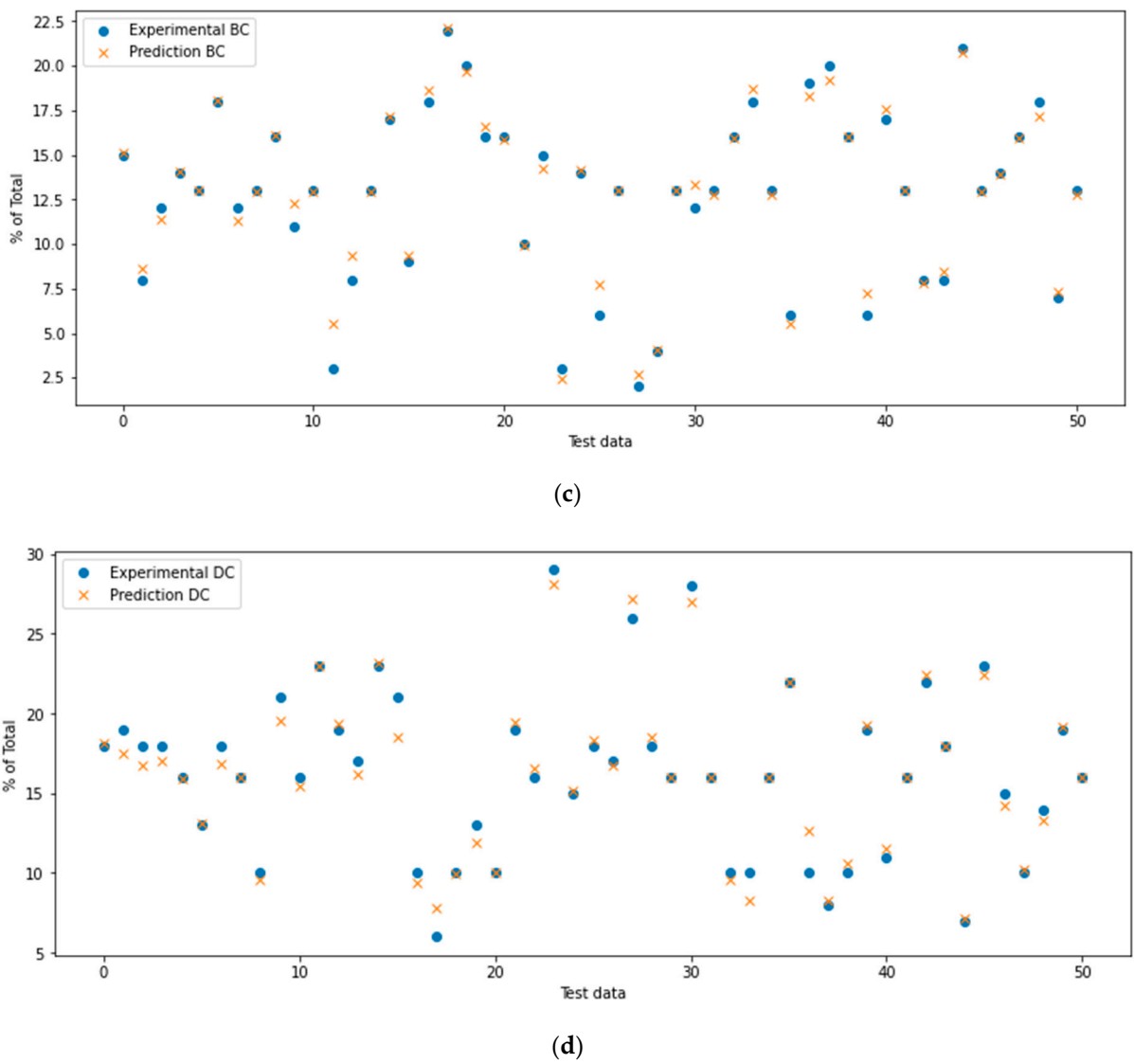

**Figure 5.** (**a**) ANN 1; (**b**) ANN 2; (**c**) ANN 3; (**d**) ANN 4.

### 4.2. Process Optimization

The purpose of modeling any object or process is to predict its behavior. In the context of this study, it was used to optimize the process. Optimization involves the determination of the technological parameters of the process at which the values of the output parameters will be the best. The best values of CWG and BC are their maximum values, and the best value of DC is its minimum value. Thus, the problem was reduced to solving a multi-criteria optimization problem, where it is required to find two maximum and one minimum values. The problem was solved by an exhaustive search of possible values.

For this purpose, a dataset of 810 records was compiled, where the suspending medium took the values of "water", "beer wort", and "young beer"; the zeolite-containing tuff took the values of "kholinskiy zeolite", "shivyrtuin", and "pegasin"; the dose of zeolite-containing tuff varied from 0 to 4.5% with the step equaling 0.5; the storage duration varied from 0 to 4 with the step equaling 0.5.

The choice of the dose range for the application of natural minerals is determined both by their multiform effect on living objects (positive or negative, depending on the chemical nature, structure and composition of minerals), and by the hygienic safety parameters, focused on minimizing the doses of additives introduced into the product, which is also important for minimizing the production costs.

The values of the storage duration interval depend on the following factors. At the brewery, after the end of the wort fermentation process, seed yeast can immediately be introduced into the next fermentation cycle ("from tank to tank" technology); however, in the case of the unstable operation of the plant and interruptions in the functioning of the brewery, the recommended duration of yeast culture storage (2 days) may be exceeded.

The dataset was input into all four networks, then the optimal technological parameters were determined according to the best output values, and they coincided for almost all networks.

ANN 1 showed that CWG would reach the maximum values with the following input parameters: suspending medium—"beer wort", zeolite-containing tuff—"kholinskiy zeolite", tuff dose—4.5%, storage duration—4 days. BC would reach the maximum values with the following input parameters: suspending medium—"beer wort", zeolite-containing tuff—"shivyrtuin", tuff dose—4.5%, storage duration—4 days. DC would reach the minimum values with the following input parameters: suspending medium—"beer wort", zeolite-containing tuff—"kholinskiy zeolite", tuff dose—4.5%, storage duration—1 day.

ANN 2 showed that CWG would reach the maximum values with the following input parameters: suspending medium—"beer wort", zeolite-containing tuff—"kholinskiy zeolite", tuff dose—4.5%, storage duration—4 days.

ANN 3 showed that BC would reach the maximum values with the following input parameters: suspending medium—"beer wort", zeolite-containing tuff—"shivyrtuin", tuff dose—4.5%, storage duration—4 days.

ANN 4 showed that DC would reach the minimum values with the following input parameters: suspending medium—"beer wort", zeolite-containing tuff—"kholinskiy zeolite", tuff dose—4.5%, storage duration—1 day.

Thus, the four ANNs solved the optimization task in almost the same way (except storage duration for the optimal DC value in ANN 1 and ANN 4), which proves the high accuracy of the obtained models.

The obtained results allowed us to identify the following major patterns.

A positive trend in the change in physiological parameters of the yeast population during storage in the presence of natural ZT was observed throughout the duration of incubation. The effectiveness of the stimulation of the determined characteristics of yeast is expressed to a greater extent when the biomass is stored in a medium with a complete absence or lack of nutrients (water, young beer). Shivyrtuin and kholinskiy zeolite ZTs showed higher bioactivity compared to the tuff from the Pegas River deposit.

The chemical composition and properties of natural ZTs determine the causes of the above phenomena [14,15]. Mineral elements are present in ZT in forms that are easily assimilated by biological objects. The main component of tuffs—silicon—has a priority impact on living organisms including yeast. The exchange ions (sodium, potassium, calcium) and the impurities of magnesium, zinc, iron, manganese, copper, etc. contained in a small amount in the tuffs also have a major impact on living organisms [4,33–36]. Unlike shivyrtuin and kholinskiy zeolite ZTs, natural pegasin contains more impurities of iron, tin, molybdenum, etc., which are harmful to yeast, which perhaps explains the lesser degree of its influence on the variability of the number of budding cells, cells with glycogen, and dead cells in the yeast culture during storage.

Another possible aspect of the beneficial effect of ZTs on microbial culture is the ability of minerals to absorb the toxic substances both entering the storage medium from the outside and released by the yeast itself.

The most favorable medium for storage is wort, as it contains all the substances necessary for yeast.

If the duration of seed yeast incubation needs to be increased, the introduction of natural minerals into the suspending medium helps maintain the normal physiological state of the microbial culture, with a high level of budding cells and cells with glycogen, and a low concentration of dead cells.

Exceeding the shelf life by more than 4 days will negatively affect the biotechnological parameters of the yeast population, regardless of the studied media used (even if it is a valuable medium such as beer wort): after the available nutrients have been consumed (in wort or in young beer) or since they are initially absent (in water), the glycogen contained in yeast cells is consumed, which leads to a decrease in the vital activity of the culture and the increase in the number of dead cells. All of this would finally slow down the process of alcoholic fermentation, worsen the taste and aroma properties of beer, and reduce the colloidal resistance of the finished drink.

To prevent intensive cell death, the duration of yeast culture storage under a layer of beer wort with the introduction of kholinskiy zeolite-containing tuff should be no more than 2 days. The optimal dose of minerals (kholinskiy zeolite and shivyrtuin, which would ensure the maximum accumulation of cells with glycogen and budding yeast cells (66–69% and 20–22% of the total respectively) and the minimum content of dead cells in the yeast population (5–6% of the total) under the given conditions was 4.5% of the suspending medium volume.

## 5. Conclusions

Four fully-connected neural networks were designed. These are capable of predicting the number of cells with glycogen, the number of budding cells, and the number of dead cells in the yeast stored in various suspending media using various zeolite-containing tuffs as a stimulating additive at different dosages and different storage duration.

The accuracy of each ANN was measured using the mean relative error. All networks had a low margin of error: ANN 1—4.869%; ANN 2—1.8381%; ANN 3—6.2905%; ANN 4—4.2191%. Such errors are acceptable for technological processes, which means that the models are adequate and can be used.

With the help of ANNs, the optimal parameters for yeast storage with natural minerals before introduction into the fermentation medium were determined. To maximize the amount of cells with glycogen (CWG) in the yeast culture (66–69% of the total), the storage process should be run at the following input parameters: suspending medium—beer wort, zeolite-containing tuff–kholinskiy zeolite, tuff dose—4.5%, storage duration—4 days. The amount of budding cells (BC) can be maximized (20–22% of the total) when the yeast is stored in beer wort for 4 days with 4.5% of shivyrtuin tuff added to the storage medium. The minimum content of dead yeast cells (DC) (5–6% of the total) was achieved by suspending the yeast in the beer wort for 2 days with kholinskiy zeolite tuff.

It was established that ANNs can be used for mathematical modeling of the process of correcting the unfavorable transformations of the biotechnological properties of brewer's seed yeast at the storage stage using natural zeolite-containing tuffs.

In the future, it is planned to investigate the use of ANNs for modeling other technological processes in food science technology.

**Author Contributions:** A.V.S.—creation of neural networks, writing the manuscript; L.V.P.—analytical review of the literature, organization of experimental research, obtaining factual material; D.M.B.—research organization, final approval of the version to be submitted; I.Y.S.—research methodology, design of the study. All authors have read and agreed to the published version of the manuscript.

**Funding:** The research was carried out through a grant of the Russian Science Foundation No. 22-26-20102, https://rscf.ru/project/22-26-20102/ (accessed on 1 November 2021) and from the Kemerovo region (Kuzbass).

**Institutional Review Board Statement:** Not applicable.

**Informed Consent Statement:** Not applicable.

**Data Availability Statement:** Data are available from the authors upon request.

**Conflicts of Interest:** The authors declare that they have no known competing financial interests or personal relationships that could have appeared to influence the work reported in this paper.

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
