# Peer review of "Modeling the Physiological Parameters of Brewer’s Yeast during Storage with Natural Zeolite-Containing Tuffs Using Artificial Neural Networks"

_information, doi:10.3390/info13110529_

Round 1
Reviewer 1 Report
Dear all,
below are my comments and suggestions
Manuscript ID: information-1964338
Materials and Methods. It was difficult because the text is not understandable in some places. Please verify the method description.
Please check the entire manuscript, and write in italics all scientific names (e.g. Line 90)
Figures 1,2,3 are difficult to understand, please use another size
Please check the citations and References in the main manuscript.
According to the Instructions for Authors, in the text, reference numbers should be placed in square brackets [ ], and placed before the punctuation; for example [1], [1–3] or [1,3]. The writing of the manuscript is not neat.
The Conclusions section can be improved and include more relevant results.
The number of bibliographic sources is adequate, more than 70% of the total bibliographic sources are from the last 5 years.
There are some grammatical errors and instances of badly worded/constructed sentences throughout the manuscript. Please refine the language carefully.
Author Response
Point 1: Materials and Methods. It was difficult because the text is not understandable in some places. Please verify the method description.
Response 1: Clarifications have been made to clause 2.1. according to the materials used, in clause 2.2. a detailed description of methods for studying yeasts is given.
Point 2: Please check the entire manuscript, and write in italics all scientific names (e.g. Line 90)
Response 2: The generic and specific name of the yeast is in italics and is indicated in Latin.
Point 3: Figures 1,2,3 are difficult to understand, please use another size
Response 3: Figures 1, 2, 3 (clause 3.1.) are presented in a different format, with an indication of the confidence interval, the inscriptions on the figures have been corrected.
Point 4: Please check the citations and References in the main manuscript.
Point 5: According to the Instructions for Authors, in the text, reference numbers should be placed in square brackets [ ], and placed before the punctuation; for example [1], [1–3] or [1,3]. The writing of the manuscript is not neat.
Response 4&5: References to the literature in the text and the list of references have been corrected in accordance with the requirements.
Point 6: The Conclusions section can be improved and include more relevant results.
Response 6: The conclusions have been corrected (specific figures are presented).
Point 7: The number of bibliographic sources is adequate, more than 70% of the total bibliographic sources are from the last 5 years.
Response 7: Without changes
Point 8: There are some grammatical errors and instances of badly worded/constructed sentences throughout the manuscript. Please refine the language carefully.
Response 8: The text was checked for grammatical and lexical errors, corrections were made.
Reviewer 2 Report
1. Please add more results to the abstract.
Line: 90The names of microorganism should be in italics.
2. There is no information on how the wort was prepared for testing, what malt and what hops. What content of alpha acids, as they affect the work of yeast during fermentation process.
3. Please provide a detailed methodology for determining the viability and number of yeast cells.
4. Please correct figure 1 to make it more readable. Are standard deviations taken into account?
5. This applies to all figures.
Author Response
Point 1: Please add more results to the abstract. Line: 90The names of microorganism should be in italics.
Response 1: Abstract corrected, more specific information added. The generic and specific names of yeasts are in italics and are indicated in Latin (corrected).
Point 2: There is no information on how the wort was prepared for testing, what malt and what hops. What content of alpha acids, as they affect the work of yeast during fermentation process.
Response 2: In clause 2.1. added raw materials used for the preparation of wort and its quality indicators
Point 3: Please provide a detailed methodology for determining the viability and number of yeast cells.
Response 3: In clause 2.2. added a detailed description of yeast research methods
Point 4: Please correct figure 1 to make it more readable. Are standard deviations taken into account?
Response 4: Figures 1, 2, 3 (clause 3.1.) are presented in a different format, with an indication of the confidence interval, the inscriptions on the figures have been corrected
Point 5: This applies to all figures.
Response 5: Figures corrected.
Round 2
Reviewer 1 Report
I recommend publishing the manuscript in this form.